# HealthyLIFE, a Combined Lifestyle Intervention for Overweight and Obese Adults: A Descriptive Case Series Study

**DOI:** 10.3390/ijerph182211861

**Published:** 2021-11-12

**Authors:** Nicole Philippens, Ester Janssen, Sacha Verjans-Janssen, Stef Kremers, Rik Crutzen

**Affiliations:** 1Department of Health Promotion, Maastricht University, 6211 LK Maastricht, The Netherlands; n.philippens@maastrichtuniversity.nl (N.P.); s.kremers@maastrichtuniversity.nl (S.K.); rik.crutzen@maastrichtuniversity.nl (R.C.); 2Ecsplore, 6163 HR Geleen, The Netherlands; srb.janssen@gmail.com

**Keywords:** lifestyle, CooL, Positive Health, overweight, obesity, holistic

## Abstract

(1) Background: The aim of this study is to investigate changes over time in participants of healthyLIFE, a Combined Lifestyle Intervention (CLI) based on the Coaching on Lifestyle (CooL) intervention. This study focuses on changes in behavior, physical fitness, motivation and Positive Health eight months after the start of the intervention. (2) Methods: In total, 602 Dutch adults, meeting the CLI inclusion criteria, were included from January 2018 until October 2020 in this descriptive case series study. We collected a broad set of data regarding weight/BMI, physical fitness, motivation, self-efficacy, social influence, personal barriers and needs towards food and physical activity and perceived personal health by means of the six dimensions of Positive Health. (3) Results: Eight months after baseline, positive effects of the intervention were found on most outcome measures. We found an increase in all measured aspects of physical fitness (stamina, flexibility, mobility, hand grip strength and BMI). Dietary changes were limited during the healthyLIFE intervention, except for fruit consumption (increase with an effect size of 0.42). The largest effect sizes were found for the dimensions of Positive Health ranging from 0.41 to 0.68. (4) Conclusion: The healthyLIFE intervention is successful in improving participants’ BMI, physical fitness, and perceived physical, mental and social health. A broad health perspective, beyond physical measurements, is recommended when studying effects of the CLI.

## 1. Introduction

In 2020, 50.0% of Dutch people aged 18 and older were overweight and 13.9% obese [1]. Overweight is more common in men, while obesity is more common in women. Obesity is linked to many diseases, such as type 2 diabetes mellitus, cardiovascular disease and various cancers [2,3,4]. Obesity is also linked to mental health problems: obese persons have a 55% increased risk of developing depression over time [5], whereas the association between obesity and mental health directly impacts quality of life via various pathways [6].

Other conditions associated with overweight or obesity are respiratory diseases, such as asthma, and diseases of the musculoskeletal system, such as chronic back pain and osteoarthritis [3,7]. Obesity or severe obesity can lead to infertility in both men and women [8,9]. In addition, being overweight in pregnant women increases the risk of miscarriage [2,8]. A study during the recent COVID-19 pandemic demonstrated that obesity itself is considered a risk factor for COVID infection and mortality [10].

### 1.1. Combined Lifestyle Interventions

As of January 2019, Combined Lifestyle Interventions (CLIs) are part of basic health insurance in the Netherlands. A CLI is a health care intervention for people with overweight or obesity. A CLI promotes healthy lifestyle changes by focusing on behavior change, resulting in weight loss. Being part of basic health insurance policy makes the CLI easily accessible for the target population.

CLIs need to be approved by the Dutch Institute for Public Health and Environment (in Dutch: RIVM) to be eligible for inclusion in basic health insurance policy. Approval is only available for CLIs that have proven to be effective or at least have shown initial signs of effectiveness in facilitating weight reduction. At this moment, four CLIs are approved: SLIMMER, Beweegkuur, Samen Sportief in Beweging (SSIB) and CooL. All CLIs stimulate weight reduction by promoting sustained healthier behavior and consist of a treatment and a maintenance phase, covering 24 months in total in which participants are coached towards a healthier lifestyle. The CLIs exist of a combination of group and individual sessions and cover at least the topics of healthy diet, physical activity and behavioral change necessary to support a sustained lifestyle change. SLIMMER, Beweegkuur and SSIB are executed by a team consisting of a physiotherapist, dietician and lifestyle coach. The CooL intervention is different in the sense that there is only one lifestyle coach who covers all topics and sessions, that all lifestyle topics (e.g., physical activity, diet, sleep, stress, relaxation, time management, and planning) are considered potentially equally important, that it is based on an autonomy-supportive coaching style and that the content of the intervention has an open character.

The inclusion criteria for all CLIs are: (1) being 18 or older; (2a) having a Body Mass Index (BMI) between 25 and 30 kg/m^2^ in combination with a large waist circumference (>88 cm for women and >102 cm for men) or with comorbidity ((increased risk of) diabetes or cardiovascular disease, osteoarthritis or sleep apnoea), or (2b) having a BMI >30 kg/m^2^ regardless of waist size or comorbidity; and (3) being sufficiently motivated to complete the two-year intervention as judged by the referrer, e.g., the general practitioner or practice nurse, and lifestyle coach.

Pilot studies in the Netherlands have shown positive effects of four CLIs on health-related outcomes, such as weight, BMI, waist circumference, fasting insulin and blood glucose [11,12,13,14]. However, health includes more than biomedical aspects. Huber et al. describes health as the ability to adapt and to self-manage, in the face of social, physical and emotional challenges, also known as the concept of Positive Health [15].

### 1.2. Positive Health

Positive Health represents a holistic view on health, expressed by six dimensions consisting of various underlying aspects that are all indicators for health. The dimensions of Positive Health exist of bodily functions, mental well-being, meaningfulness, quality of life, participation and daily functioning. Positive Health focuses on a person’s resilience and self-management instead of one’s disease or disabilities. Having this broad perspective on a person’s health enables people to deal with the physical, emotional and social challenges in life and to be in charge of their own life. The six dimensions are displayed in a spider web with six axes ranging from value 0 (in the centre, for poor) to 10 (on the periphery, for excellent) and together make up My Positive Health, enabling a dialogue between patient and health care provider on the broad perspective of one’s personal health [15].

### 1.3. Our Approach

The concept of Positive Health supports the idea to monitor changes in CLI participants over time from a broad health perspective. In other words, by not only focusing on physical results using physical tests and measurements, but also incorporating self-reported outcomes such as quality of life, motivation and personal beliefs. This will provide us more insight into the exact mechanisms of CLIs and their effects on health from the perspective of Positive Health and the relation with overweight and obesity in the Netherlands. The aim of the current study is to investigate changes over time, including changes in the dimensions of Positive Health in participants of healthyLIFE, a CLI based on the Coaching on Lifestyle (CooL) intervention. This study focuses on changes in behavior, physical fitness, motivation and Positive Health after 14 weeks and 8 months of the start of the intervention.

## 2. Materials and Methods

### 2.1. CooL: A Combined Lifestyle Intervention

Access to CLIs should be possible upon request for everyone with a Dutch health insurance policy, given that he or she meets the inclusion criteria. However, this right to basic health care limits the study design options to investigate effectiveness of CLIs; conducting a randomized clinical trial would be unethical. Since the CLI is not yet widely marketed there are hardly any waiting lists that could act as an alternative source of control group members. Hence, this study uses a pre-post-test design to assess effects of a CLI in daily practice (i.e., a descriptive case series study).

The CooL intervention consists of group and individual sessions addressing a range of lifestyle-related topics among which physical activity, dietary behavior, sleep and stress. The CooL intervention aims for higher perceived quality of life, healthier eating habits (including more fruit and vegetable intake, less sugar and snack consumption, more regularity), more physical activity, less sedentary behavior, more investment in good sleep and relaxation, and positive changes in physical outcomes such as weight, BMI and waist circumference. The intervention consists of a treatment program of 8 months followed by a maintenance program of 16 months, including an intake (1 h), 16 group sessions (each 1.5 h) divided over 2 years, and individual sessions (6 h in total) divided over two years [11].

The CooL intervention is an open CLI, which means that lifestyle coaches adapt the intervention to their target audience, within certain boundaries and restrictions. However, a predefined set of final objectives of the CooL intervention are pursued and the main effective elements (e.g., goal setting, mobilizing social support, Positive Health and positive psychology, self-management and self-monitoring) of the CooL intervention are respected [11]. The lifestyle coach encourages participants to take responsibility for their personal lifestyle changes. The lifestyle coach is a trained and licensed professional who coaches the participants in identifying and mapping personal health-related behavior. The lifestyle coach addresses motivation, personal objectives and behavioral change. The main objective is to coach and enable participants to adhere to a sustained healthier lifestyle in line with the individual needs and personal goals of the participants.

### 2.2. HealthyLIFE

HealthyLIFE consists of a lifestyle intervention and a physical activity program. The lifestyle intervention is based on the CooL intervention. Participants of the healthyLIFE intervention are coached by a lifestyle coach and a physical activity coach. The intervention takes two years. During the first fourteen weeks of the intervention, participants take part in the physical activity program. Participants exercise on a weekly basis in small groups of approximately eight people. The aim of the physical activity program is to improve participants’ physical fitness and stimulate them to stay physically active after the end of the intervention. The physical activity coach introduces different forms of physical exercise and coaches the participants on finding suitable activities, which can be done on a regular basis. These activities can be performed unorganized, for example walking or cycling, or organized, for example as a member of a local sports club or seniors’ gym. The healthyLIFE intervention integrates the concept of Positive Health from the Institute for Positive Health (iPH) by using the Positive Health dialogue tool during intake, after completing the first part of the lifestyle intervention (8 months) and at the end of the intervention (2 years). Two of the sixteen group sessions of the lifestyle intervention are dedicated to Positive Health. Though Positive Health is integrated in the regular working method of the lifestyle coach, all healthyLIFE coaches receive an additional Positive Health training to revive their knowledge and perspective on this concept.

### 2.3. Design and Study Population

This study is a descriptive case series study in which participants of the healthyLIFE intervention were monitored on four different occasions over time. The participants, all Dutch-speaking adults living in the Southern part of the Netherlands, were included from January 2018 until October 2020. All participants met the inclusion criteria for participating in a CLI. Participants were referred to the healthyLIFE intervention by their general practitioner or practice nurse. The decision on a proper fit for inclusion was up to the participant, the referrer and the lifestyle coach. The study participants signed an informed consent regarding data collection for this study.

### 2.4. Data Collection

We used anthropometric measurements, a physical fitness test and a lifestyle questionnaire to collect a broad set of data. The lifestyle questionnaire was based on existing validated questionnaires and to a large extent similar to the questionnaires used during the pilot phase of the CooL intervention [16]. The outcome measures we collected consisted of weight/BMI, physical fitness, motivation, self-efficacy, social influence, personal barriers and needs regarding food and physical activity. In addition, we collected data on a person’s health by means of the online iPH questionnaire on the six dimensions of Positive Health.

Data were collected at four time points during the healthyLIFE intervention: at the beginning of the intervention, during the intake (T0); after 14 weeks, at completion of the physical activity program (T1); after 32 weeks, at completion of the first part of the lifestyle program (T2); and after 24 months, at completion of the intervention (T3). The data from T3 were not yet available at the time of the analysis.

### 2.5. Demographic Characteristics

At baseline, participants were asked to report their personal characteristics such as gender, date of birth, country of birth and highest complete education, marital status, living situation and occupational status. We subdivided educational level into the categories: low (i.e., no education or primary education), intermediate (e.g., secondary education), and high (e.g., tertiary education). The living situation is divided into living together with someone (married or cohabiting) and living alone (divorced, unmarried or widowed). The occupational status is also divided into two categories: working (paid work, voluntary work or self-employed) and not working (homemaker, unemployed/job seeker, retired/in early retirement, disabled or student).

### 2.6. Weight and BMI

Anthropometric data were collected by using a Seca stadiometer for height (Seca 217, Hamburg, Germany) and a Seca weighing scale (Seca 899, Hamburg, Germany) for measuring weight.

### 2.7. Physical Fitness

Data regarding physical fitness were collected by executing the GALM physical fitness test for elderly made up of four different physical tests: (1) mobility: Timed Up and Go (TUG)—the time that a person takes to rise from a chair, walk three m, turn around 180°, walk back to the chair, and sit down while turning 180°; (2) stamina: 6 min Walk—walking distance covered over a time of 6 min compared to reference values based on age and gender; (3) flexibility: flexibility of the lower back and hamstring muscles using the Sit and Reach box; (4) hand grip strength: squeeze strength in kilogram or Newton using a hand dynamometer compared to reference values based on age and gender [17].

### 2.8. Motivation to Be Physically Active

We used the Behavioral Regulation in Exercise Questionnaire (BREQ-3) to collect data on motivation regarding physical activity [18,19]. The BREQ-3 is a multi-dimensional questionnaire to examine motivation to be physically active based on the assumption that motivation varies along a continuum of perceived self-determination ranging from non-self-determined (or controlled) to self-determined (or autonomous) forms of behavioral regulation [20].

Two constructs were derived from the BREQ data: firstly, autonomous motivation, the extent to which exercise behavior is regulated on a voluntary basis, a feeling of free choice, and secondly, the counterpart being controlled motivation, the extent to which exercise behavior is regulated by the desire to please others, receive rewards, or avoid negative reactions and emotions. Questions were answered on a five-point Likert scale ranging from ‘totally disagree’ to ‘totally agree’.

### 2.9. Personal Needs and Satisfaction on Physical Activity

We used the Psychological Need Satisfaction in Exercise questionnaire (PNSE) to collect data on personal needs on physical activity. The PNSE is a multi-dimensional questionnaire to examine the perceived psychological needs and satisfaction related to physical activity based on the Self-Determination Theory [21]. We used three constructs to summarize the data on the PNSE questionnaire—PNSE autonomy: the extent of freedom experienced in choosing and shaping one’s own physical activities; PNSE relatedness: the extent of social interaction and attachment experienced during physical activities; PNSE competence: the extent of self-confidence and capabilities experienced to carry out challenging physical activities. Questions were answered on a five-point Likert scale ranging from ‘totally disagree’ to ‘totally agree’.

### 2.10. Barriers

Data on perceived barriers were collected by questions on personal, physical, social and mental barriers perceived by the participant to participate in physical activities, for example ‘My health is not good enough’ or ‘I’m (often) too tired to move’. Similar questions were used to examine the barriers for healthy eating: ranging from the cost of healthy food to the lack of tasty recipes. Questions were answered on a five-point Likert scale ranging from ‘totally disagree’ to ‘totally agree.’

### 2.11. Self-Efficacy

To investigate self-efficacy, we used questions to examine to which extent a participant considered himself or herself capable of exercising more and eating healthier, despite personal and environmental factors (for example being tired, running out of time, lacking social support) that exert a negative influence. Questions were answered on a five-point Likert scale ranging from ‘totally disagree’ to ‘totally agree.’

### 2.12. Social Influence

We used several questions to assess social support and social pressure in physical activity. Questions on social support covered the extent to which the participant felt positively supported in initiating and carrying out an exercise. Examples of social support were encouragement, supporting attitude, direct help or making compliments. Social pressure in physical activity consisted of questions on the extent the participant felt negatively supported in initiating and carrying out exercise activities. Examples of social pressure were discouragement, criticism or being laughed at. Questions were answered on a five-point Likert scale ranging from ‘never’ to ‘very often.’

### 2.13. Dietary Behavior

Dietary behavior was measured using a food frequency questionnaire. Participants were asked how often and how much of the following items were consumed during a regular week: sweet snacks (e.g., cake, candy bars, chocolate, and cookies), salty snacks (e.g., fried food, nuts, potato chips, and cheese), takeaway meals (e.g., fried food, Chinese food, and pizza), cooked vegetables, salads, and raw vegetables. We clustered these items into four main categories: (1) snacks: a sum of the total number of times that people took sweet and/or salty snacks on a weekly basis, (2) unhealthy food: a sum of the total number of times that people consumed a takeaway meal (fried, Asian, Greek and/or Dutch) on a weekly basis, (3) fruit: a sum of the total number of pieces of fruit that people took on a weekly basis and (4) vegetables and salad: a sum of the total number of spoons of vegetables, salad and raw vegetables that people consumed on a weekly basis.

### 2.14. Positive Health

The data for Positive Health were collected via an online questionnaire [22]. The Positive Health measurements were carried out at the start of the intervention (T0) and at T2 (after 8 months). The dataset consisted of the responses to 17 items of the 42-item questionnaire. This 17-item measuring instrument was validated by Van Vliet et al. [23]. The 17 items were combined into six factors, i.e., physical fitness (feeling healthy, feeling fit, being physically active), mental functions (being able to remember and being able to concentrate), future perspective (being able to deal with changes, striving for ambitions, and feeling confident towards one’s own future), contentment (being happy, feeling well and feeling well balanced), social relations (social relations, social support, and sense of belonging) and daily life management (knowing one’s limitations, health knowledge, and time management).

### 2.15. Statistical Analyses

As a preparatory step, we performed a factor analysis followed by calculating omega to assess the internal structure of items regarding constructs such as social influence, self-efficacy, motivation and psychological needs, for example BREQ and PNSE [24]. This factor analysis justified summarizing all lifestyle constructs by item score means. Missing data were excluded from the statistical analyses. We used a similar approach for Positive Health in line with Van Vliet et al. [23]. We averaged the item scores per factor for all available data on T0 and T2: physical, Mental functioning, future perspective, contentment, social relations and daily life management. Again, the factor analysis followed by calculating omega to assess the internal structure justified summarizing of the constructs by item score means. All factor analyses were performed using R software.

Descriptive statistics were performed. Changes over time were analyzed using paired T-tests (T0 versus T1 and T0 versus T2) for all constructs. All paired T-tests were performed using SPSS software (version 25). Only cases with complete data (at T0 and T1, or at T0 and T2) were included in the analysis per comparison. For each comparison, we calculated Cohen’s d as a standardized effect size allowing comparison between constructs. Cohen’s D was calculated using the online calculator available at https://memory.psych.mun.ca/models/stats/effect_size.shtml (accessed on 5 November 2021). An effect size smaller than 0.20 is considered very small, an effect size between 0.20 and 0.50 is considered small, an effect size between 0.50 and 0.80 is considered medium and an effect size greater than 0.80 is considered large [25].

### 2.16. Ethics

This study was registered in the Dutch Trial Register (NTR7018, registration number 17N174) and it was exempt from review by a research ethics committee as it does not fall within the scope of the Dutch Medical Research Involving Human Subjects Act Central Committee on Research Involving Human Subjects [26]. All participants gave their informed consent for their pseudonymized personal data to be used for research purposes.

## 3. Results

### 3.1. Participants

The baseline tests and questionnaires on lifestyle, physical fitness and Positive Health were taken from 602 participating adults between April 2018 and October 2020. For each subset, the amount of missing data differed per item per measurement, except for the Positive Health subset. This is a subset of 88 participants who completed the entire online questionnaire on Positive Health both on T0 and T2. In Figure 1, an overview of the different data subsets with their respective N on the three measurements is displayed.

Of all participants, a total of 39% were male and 61% female. Most participants (90%) had a Dutch background. Approximately two-thirds of the participants had a lower or intermediate level of education; 40–50% did not have a steady job (anymore); and over 70% of the participants were living together with a partner (Table 1).

### 3.2. BMI and Physical Fitness

The average BMI of the participants at T0 was 34.45 kg/m^2^ (Table 2). The BMI of the participants improved at T1 and T2. The effect size of the change in BMI was small at T1 and increased to medium at T2. On average, the participants showed a decrease in body weight at T2 of 2.44 kg, corresponding to a 2.5% average weight loss per participant and an average decrease in BMI of 0.85. Of the participants, 22% had a weight loss of 5% or more at T2 compared to T0.

All physical tests (BMI, stamina, mobility, flexibility and handgrip strength) improved over time both on T1 and T2. At T1, the effect sizes were small on stamina and handgrip strength and very small on flexibility and the TUG test. The positive effect was still existent at T2; the effect size increased to medium for stamina and small for all other physical tests (Table 2).

### 3.3. Lifestyle Questionnaire

#### 3.3.1. Physical Activity-Related Factors

The autonomous motivation (BREQ Autonomy) to participate in physical activity increased significantly over time (ES = 0.44 at T1 and ES = 0.50 at T2) (Table 3). The controlled motivation (BREQ Control) to participate in physical activity significantly decreased at T1 (ES = 0.25). Since the BREQ Control construct is calculated by multiplying the underlying items with a weighted negative factor, the result in time shows a positive difference whereas in fact it is a decrease [19]. This effect diminished at T2.

Perceived needs satisfaction regarding autonomy, relatedness, and competence in physical activity increased from T0 to T1. Effect sizes were small at T1. The positive effect on perceived autonomy was also visible at T2. Participants experienced less barriers to be physical active at T1. The effect size was small. Participants experienced more social support on physical activity at T1; the effect size was small. There were no significant effects on self-efficacy to exercise and on social pressure to be physically active.

#### 3.3.2. Dietary Behavior and Dietary Behavior-Related Factors

The changes over time on snacking and fried food/takeaway were limited at T1 and T2 (Table 4). The effect size on vegetable intake was small at T1 and no longer significant at T2. There was a small positive change on fruit consumption at T1, showing an increase to a small effect size at T2. In practice, the participants increased their fruit intake on average with 2.6 pieces of fruit more per week at T2. Self-efficacy in relation to dietary behavior decreased at T1 with a very small effect size, whereas, at T2, this change was no longer significant. There was no change over time on the perceived barriers in relation to dietary behavior at both measurements during the intervention.

### 3.4. Positive Health

All six factors of Positive Health increased (Table 5). The standardized effect size showed a small effect on mental functioning and future perspective of the participants and a medium effect on contentment, social relations, daily life management and physical fitness with effect sizes ranging from 0.41 to 0.68. In other words, within the time frame of the first eight months of the intervention, participants experienced improvements in remembering and concentrating (mental functions). They felt more confident in dealing with the changes ahead striving for ambitions (future perspective). They denoted more happiness and felt more balanced (contentment), enjoyed more social support and sense of belonging (social relations) and felt more skilled to handle daily life (daily life management). The largest effect (medium effect) was realized on physical fitness: participants felt healthier, more fit and more physical active compared to baseline.

## 4. Discussion

### 4.1. Changes over Time of HealthyLIFE

The present study examined longitudinal changes in behavior, physical fitness, quality of life, personal beliefs and motivation with regard to the lifestyle and personal health of participants in the healthyLIFE intervention. Overall, positive effects of the intervention were found on most outcome measures. The largest effect sizes were found for the dimensions of Positive Health.

This is the first study to show effects of a Combined Lifestyle Intervention on Positive Health. In short, participants showed an increase on all Positive Health factors after the first eight months of the healthyLIFE intervention, meaning that besides weight reduction, the participants had a more positive perspective on their personal and social health and well-being.

The fact that all these aspects of health are covered by the intervention and monitored during the change process is important: as a review by Warkentin illustrated, a decreased BMI improves physical health but not necessarily mental health [27]. In fact, the opposite might be more evident: mood and quality of life predict the effect on weight loss [28].

The identified changes over time on all Positive Health dimensions might be explained by healthyLIFE’s strong focus on increasing self-management. Self-management is an important aspect of weight control and as such a crucial pillar of the CooL intervention. Interventions aiming at self-control have shown beneficial effects in dietary behavior, physical activity and weight loss [29]. Self-management, or self-regulation, consists of setting important challenging goals, striving towards your goals and dealing with the challenges that you come across [30]. It has been shown that the effect of self-management interventions based on pro-active coping can be sustained until nine months after the intervention. Pro-active coping encompasses anticipation planning and evaluation of self-management [31].

Participants of healthyLIFE are encouraged to take responsibility for their personal lifestyle changes as part of the regular working method of the lifestyle coach. Furthermore, the concept of self-management is a crucial part of the model of Positive Health. This model includes the concept of self-management not only by stimulating behavior related to self-management but also by addressing a supporting attitude related to self-management. This self-management attitude is defined as taking one’s own direction and being able and self-confident to do so [32].

In addition to psychological mechanisms on well-being, several metabolic mechanisms can shed light on the effect of the (behavioral) lifestyle changes and the outcomes on well-being. Physical activity modulates several neurotransmitters associated with alertness, pleasure and reward, and the level of anxiety. Other neurochemical factors, such as opioids and endocannabinoids, may be released during physical activity, promoting a sense of euphoria or well-being [33]. In addition, physical activity impacts the nervous system, acting as an antidepressant or an anxiolytic, potentially improving mood, self-esteem, and cognition [34].

The relation between fruit intake and mental well-being is mainly focusing on psychological mechanisms, e.g., the belief that some foods (such as fruits and vegetables) are particularly healthy, whereas other foods (such as potato chips) are particularly unhealthy. These beliefs or expectations may give rise to feelings of virtue or self-efficacy after consuming ‘good’ foods, and feelings of guilt or lack of self-control after consuming ‘bad’ foods thereby impacting short-term mood. In addition, high fruit and vegetable intake may predict more positive subjective evaluations of physical health [35]. Functional components of food (which are found a.o. in fruit) impact health and well-being via metabolic pathways [36]. Future studies on these mechanisms would be beneficial, especially in light of the CLI.

The results of the present study add to the scientific knowledge around the concept of Positive Health and show that lifestyle coaches can support overweight or obese individuals in improving their perceived health and quality of life. Quality of life is an important outcome as it can be linked to self-efficacy and improvements in mood state and mental functioning, even when weight loss is limited [37].

Eight months after baseline, an increase in all measured aspects of physical fitness (stamina, flexibility, mobility, hand grip strength and BMI) was found. Stamina (6 min walk) and BMI improved most (medium effect size). With respect to dietary behavior, we detected an increase in fruit intake and regarding motivation we found an increase in autonomous motivation for physical activity.

The decrease of 0.85 points on BMI (corresponding with an average of 2.4 kg) after 8 months is in line with earlier studies in the Netherlands on comparable interventions such as CooL [11], SLIMMER [12] and Beweegkuur [13]. Note that this study was partly running during the time of the COVID-19 restrictions, almost one fifth of the respondents completed the intervention during COVID-19 lockdown. The preventive measures regarding COVID-19, especially working at home and self-isolation, have negatively affected the lifestyle of Dutch citizens and led in general to a more sedentary lifestyle resulting in weight gain [38]. Two recent studies indicate an average weight gain of 1.5 and 2 kg, respectively, during the COVID-19 lockdown [39,40], whereas an online questionnaire in 30 countries even indicates an average weight gain of 5.6 kilos in the Netherlands [41]. Stabilizing one’s personal weight might as well be considered a success during the COVID-19 lockdown period though the lack of a control group in our study design prevents us from drawing strong conclusions.

Regarding the increase in stamina among the healthyLIFE participants: SLIMMER reported an increase in stamina as well, whereas the increase in hand grip strength was not seen or measured in the other interventions [11,12,42]. Regarding the motivational aspects of behavior change we found an increase in autonomous motivation for physical activity with medium effect size which is similar to the CooL pilot [11]. Though the physical activity program was an extra stimulating factor, the focus on self-management might be the key to the increase in physical fitness and a shift towards more autonomous motivation for physical activity. Lifestyle coaches in healthyLIFE used a similar approach as the lifestyle coaches in the original CooL pilot: both stimulated participants to try out different physical activities and encouraged to select a physical activity that met their personal criteria. The focus for both interventions was on easy-to-access physical activity integrated within daily life whereas healthyLIFE participants had the additional benefit of an actual in-house physical activity program during the first weeks of the intervention. More research is needed to examine the effects of an additional physical activity program to the CLI.

Dietary changes were limited during the healthyLIFE intervention, except for fruit consumption. Participants increased fruit intake with on average 2.6 pieces of fruit per week, an outcome which is comparable to CooL and larger than the increase shown in Beweegkuur [11,42]. Vegetable consumption after eight months showed no significant change. This finding is in line with the effect of the SLIMMER intervention whereas CooL and Beweegkuur did show an increase in vegetable intake compared to baseline after 8 months and 1 year, respectively [11,12,42]. As the healthyLIFE intervention is based on self-management, participants decide on their personal lifestyle actions thereby stimulated to formulate sequential actions in time. Consequently, though many improvements can be made during eight months, some health topics might still be left untouched by the participants.

The self-efficacy regarding dietary behavior and physical activity decreased both on T1 and T2. A striking but not necessarily surprising effect. Earlier research has shown that negative task experience can rapidly result in a decrease in self-efficacy. In other words: when encountering obstacles in behavior change, participants may make a shift from unrealistic optimism towards a more realistic perspective on their abilities [43]. The outcome of the CooL intervention for children also showed a negative shift in self-efficacy on playing outside (easily accessible physical activity) and eating fruits (dietary improvement) during the first eight months of the intervention, bending towards a positive change after one year [11].

Earlier research on the CLI provided useful insights on the effects of the CLI, usually focused on some aspects of health depending on the hypothesis of the study. The results of this study on behavior, motivation, physical fitness, personal barriers and beliefs of the participants of healthyLIFE, provide a comprehensive overview on the effects of the CLI from different angles.

Obesity does not arise overnight but often has a long history involving a multitude of factors [44]. Consequently, the solution should be characterized by a broad and long-term personalized approach. Bearing in mind the length of time involved to reverse obesity (often life-long), it is not only the immediate result in kilograms that counts, but even more the learning process in which the participant acquires health-related skills such as making and persevering healthy choices. Many people consider good health of critical importance and are aware of the relation between their behavior and the effect on physical well-being. Still most people have difficulties in performing and maintaining health-promoting behavior [45].

This is the reason why we first recommend a holistic view on health when dealing with overweight or obesity, an approach that matches the competences of the lifestyle coach. Secondly, the focus of weight-management interventions should be on coaching the participants to develop self-management skills to support sustained lifestyle changes. Thirdly, we make an appeal to medical practitioners to look at the CLI from a broader perspective than focusing on weight loss only. We argue that an increase in quality of life, one’s perspective of personal well-being, is at least as important as weight loss; from the perspective of the participant, it is impossible and undesirable to choose between weight loss and feeling well balanced, feeling self-confident or feeling healthier. Since both perspectives are interconnected, we would expect to see more tangible long-term health benefits of CLI’s that have a broad perspective on health. The postulation underlying this hypothesis is that the positive effect on personal well-being translates into higher success rates of sustained health behavior change. Longer-term follow-up studies are needed to address this topic empirically.

The healthyLIFE intervention provided a very interesting opportunity to monitor the results of the intervention after eight months. Naturally, it is important to keep monitoring the results after two years when completing the intervention. These data will become available as the CLI is part of regular health care, thereby providing an excellent opportunity for further analysis.

### 4.2. Limitations and Strengths of the HealthyLIFE Study

As the CLI is part of basic health care, we did not have the opportunity to compare our results to a control group. Consequently, we used the terminology changes over time instead of effects as we cannot rule out interference with other factors and variables. However, we are confident in addressing these changes to the healthyLIFE intervention given the average effect size of the changes and the comparison to similar interventions. The outcomes of the first eight months of the healthyLIFE intervention already provide valuable information but further analysis is needed, after data collection on T3, to assess the end results of this intervention.

For practical reasons some constructs were not measured on the different measurement moments, for example motivational regulation, social support and social pressure on eating. This was mainly done to keep the participant burden within acceptable limits. The dataset contains participants that started quickly after the kick-off of the CLI being covered by basic health insurance (the minimal requirement for all Dutch citizens) but also participants that enrolled in the intervention just a few weeks or months ago. Obviously, for these participants data of follow-up measurements were not yet available.

During analyzing we used different samples for T0 versus T1 and T0 versus T2 thereby enabling a subset as large as possible from the complete dataset. For Positive Health, we used a smaller dataset since we restricted the subset to fully completed questionnaires on both T0 and T2. On the other hand, the (percentual) similarities in demographics between the subset and the overall dataset (see Appendix A, Table A1) can be classified as a strength of this study.

In future, we would like to provide a sub-group analysis on a larger dataset, which may be expected given the fact that the CLI is part of regular health care and data collection is ongoing. Continuous guidance and information on the necessity and importance of the outcome measurements to the lifestyle coaches, during the obliged training sessions of the coaches, will help to collect more complete datasets in future. Once we have a larger dataset, the accuracy of estimates per sub-group will increase, which will allow us to draw strong conclusions that we cannot yet do.

Data on actual physical activity behavior were collected during the intervention by using accelerometry. In practice, the devices provided insufficient data for research purposes due to practical reasons such as limited availability of devices, wrong use of the device and limited wear time. For that reason, data on actual physical activity behavior were not included in this study.

At the same time, there are several notable strengths to this study. It is the first research on effects of the CLI since it is part of basic health insurance. Previous studies were performed in more or less controlled circumstances while this study is based on a real life setting and thus generalizable to a wider audience. Second, the number of participants in this study is considerable compared to similar studies such as the CooL pilot study (*n* = 136) [11], SLIMMER (*n* = 516) [12] or BeweegKuur (*n* = 517) [13].

Third, we took a broad perspective on the longitudinal changes in Positive Health, behavior, physical fitness, personal beliefs and motivational regulation of participants. The data collection was based on physical tests and measurements but also on self-reported outcomes, providing us a wealth of information on changes in both physical, mental and emotional aspects.

## 5. Conclusions

The healthyLIFE intervention, a combined lifestyle intervention for overweight and obese adults, is successful in improving participants’ BMI and physical fitness. Most importantly, this study shows that the intervention also improves participants’ perceived physical, mental and social health. A broad health perspective, a perspective beyond physical measurements, is recommended when studying effects of the CLI.

## Figures and Tables

**Figure 1 ijerph-18-11861-f001:**
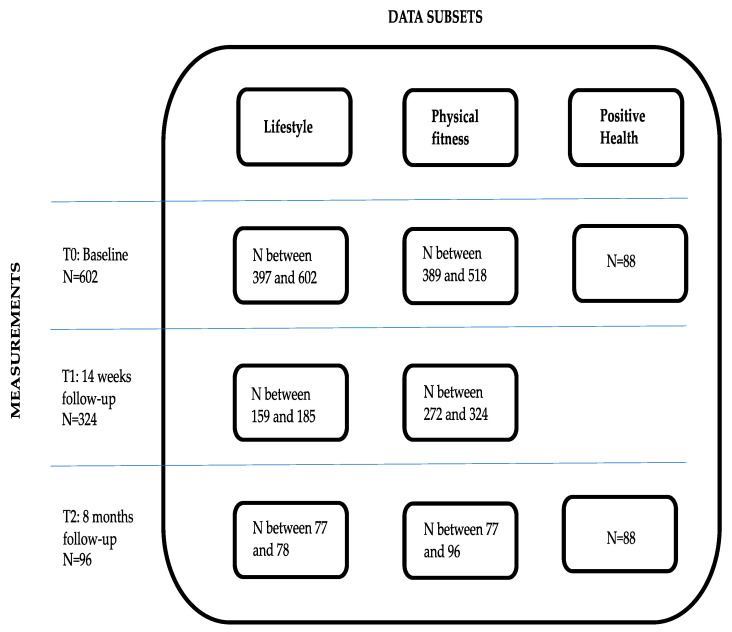
Overview of the participants on healthyLIFE.

**Table 1 ijerph-18-11861-t001:** Demographics of the participants at T0.

**Category**	**Demographic**	**Percentage of Participants in Dataset**
Gender	Male	39%
	Female	61%
Age	Until 35	3%
	35 until 45	7%
	45 until 55	24%
	55 until 65	32%
	65+	34%
Living situation	Single	28%
	Living together	72%
Land of birth	Dutch	92%
	Non-Dutch	8%
Work situation	Employed	47%
	Unemployed	53%
Education	Lower level	32%
	Medium level	43%
	Higher level	25%

**Table 2 ijerph-18-11861-t002:** Results of the physical measurements and GALM test.

Category	Construct/Factor	T0 Pre-Test (sd)	Change T0–T1 [95% CI]	Effect Size T0–T1	Change T0–T2 [95% CI]	Effect Size T0–T2
Physical measurement	BMI	34.45 (5.39)	−0.43 [−0.56, −0.30] *	0.36	−0.85 [−1.16, −0.53] *	0.54
	Weight	99.30 (18.38)	−1.26 [−1.64, −0.88] *	0.36	−2.44 [−3.36, −1.51] *	0.54
GALM test (physical fitness)	Stamina	0.78 (0.16)	0.05 [0.04, 0.07] *	0.36	0.11 [0.07, 0.15] *	0.58
	Mobility (TUG)	6.92 (2.08)	−0.33 [−0.59, −0.06] *	0.14	−0.45 [−0.78, −0.12] *	0.28
	Flexibility	21.72 (8.77)	0.90 [0.11, 1.69] *	0.14	1.27 [0.23, 2.31] *	0.28
	Handgrip strength	33.19 (11.27)	1.08 [0.50, 1.67] *	0.20	1.36 [0.15, 2.56] *	0.23

* *p* < 0.05.

**Table 3 ijerph-18-11861-t003:** Results of the self-reported measurements regarding physical activity.

Category	Construct/ Factor	T0 Pre-Test (sd)	Change T0–T1 (95% CI)	Effect Size T0–T1	Change T0–T2 (95% CI)	Effect Size T0–T2
Motivation	BREQ Autonomous	15.87 (4.74)	1.76 [1.17, 2.35] *	0.44	2.21 [1.22, 3.19] *	0.50
	BREQ Control	−5.00 (3.91)	1.02 [0.43, 1.60] *	0.25	0.87 [−0.17, 1.91]	0.19
	BREQ Total	10.84 (6.73)	2.77 [1.88, 3.66] *	0.45	3.14 [1.47, 4.80] *	0.43
Needs satisfaction	PNSE Autonomy	4.19 (0.63)	0.16 [0.05, 0.26] *	0.22	0.22 [0.07, 0.36] *	0.34
	PNSE Relatedness	3.81 (0.80)	0.32 [0.18, 0.45] *	0.37	0.17 [−0.01, 0.36]	0.21
	PNSE Competence	3.68 (0.78)	0.21 [0.08, 0.33] *	0.26	0.14 [0.00, 0.30]	0.21
Self-efficacy	Self-Efficacy to Exercise	2.51 (1.04)	−0.16 [−0.34, 0.02]	0.13	−0.29 [−0.58, 0.00]	0.23
Social	Social Support	2.04 (0.83)	0.22 [0.10, 0.35] *	0.26	0.13 [−0.08, 0.33]	0.14
	Social Pressure	1.19 (0.49)	0.06 [−0.04, 0.16]	0.09	0.07 [−0.12, 0.26]	0.08
Barriers	Barriers to Exercise	2.30 (0.64)	−0.21 [−0.29, −0.12] *	0.36	−0.16 [−0.32, 0.00]	0.22

* *p* < 0.05.

**Table 4 ijerph-18-11861-t004:** Results of the subjective measurements regarding dietary behavior.

Category	Construct/Factor	T0 Pre-Test (sd)	Change T0–T1 (95% CI)	Effect Size T0–T1	Change T0–T2 (95% CI)	Effect Size T0–T2
Dietary behavior	Salt or sweet snacking	8.09 (6.55)	−0.89 [−1.99, 0.21]	0.12	−1.71 [−3.58, 0.17]	0.21
	Takeaway or fried food	1.66 (2.87)	−0.24 [−0.69, 0.20]	0.08	−0.52 [−1.38, 0.33]	0.14
	Fruit	9.40 (7.10)	1.77 [0.64, 2.91] *	0.23	2.59 [1.18, 3.99] *	0.42
	Vegetables and salad	27.92 (16.47)	5.70 [3.31, 8.08] *	0.35	2.96 [−0.55, 6.47]	0.19
Self-efficacy	Self-efficacy for a healthy diet	2.51 (1.20)	−0.21 [−0.41, −0.01] *	0.15	−0.11 [−0.41, 0.19]	0.08
Barriers	Barriers to eat healthy	1.94 (0.73)	−0.04 [−0.13, 0.05]	0.07	−0.02 [−0.17, 0.12]	0.04

* *p* < 0.05.

**Table 5 ijerph-18-11861-t005:** Results of the Positive Health measurements.

Category	Factor	T0 Pre-Test (sd)	Change T0–T2 (95% CI)	Effect Size
Positive Health	Physical fitness	5.58 (1.78)	1.10 [0.76–1.45] *	0.68
	Mental functioning	6.28 (1.79)	0.61 [0.30–0.93] *	0.41
	Future perspective	7.09 (1.32)	0.48 [0.23–0.72] *	0.41
	Contentment	6.52 (1.55)	0.83 [0.53–1.13] *	0.58
	Social relations	7.39 (1.57)	0.61 [0.38–0.84] *	0.56
	Daily life management	7.31 (1.28)	0.67 [0.42–0.93] *	0.56

* *p* < 0.05.

## Data Availability

The data presented in this study are available on request from the corresponding author. The data are not publicly available due to ethical reasons since the informed consent statement was limited to the authors of this article.

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
