# Peer review of "HealthyLIFE, a Combined Lifestyle Intervention for Overweight and Obese Adults: A Descriptive Case Series Study"

_ijerph, 2021, doi:10.3390/ijerph182211861_

Round 1

Reviewer 1 Report

This surveillance showed that a Combined Lifestyle Intervention (CLI) based on the Coaching on Lifestyle (CooL) improved not only overweight/obesity (and related physical performance and diet behavior) but also Positive Health, an indicator of comprehensive health (physical, mental, and social health). This additive effect may be beneficial for improving the future well-being of obese people. I have some questions before publication.

Major comments

1) The sample size of the two data subsets (Lifestyle and Physical fitness) was gradually and largely decreased during surveillance. What is the reason? In addition, what are some possible ways to improve this in the future study? I wonder if the subjects who dropped out could not answer the questionnaires because the intervention was not effective for them and/or their motivation for the survey was reduced.

2) I understand that it is difficult due to the decreased sample size, but do you have any information whether gender differences, older age, work availability, and educational history influence the improving effects of the CLI on Positive Health?

Minor comments

1) 2. Materials and Methods: Please insert the subheading number to make it easier for the readers to understand.

2) In line 321-323, it is described that the controlled motivation (BREQ Control) to participate in physical activity significantly “decreased” at T1 (ES = 0.25). However, in Table 3, its value is 1.02, which is a positive value. Please check this as well other data.

Reviewer 2 Report

I reccomend a repeated measures Anova to test diff.

Reviewer 3 Report

The manuscript is well written, original and rigorous. However, according to
the reviewer's point of view, the "organic" component of the observed processes
has been completely neglected.
There were no biochemical analyzes, even non-invasive ones.
While the psychological mechanisms activated by the change in the lifestyle are
glimpsed, neither the metabolic mechanisms nor their causes are clear or
hypothesized (in terms of discussion): for example, are there any
correlations with the type of physical activity performed?
Could there be a correlation between increased fruit consumption and
improved mood, increased motivation? the positivity of thought?
Similar considerations could be made for the increased physical activity.
For the reader, these considerations may be interesting,
even if they refer to previous literature.
The overall impact of the paper could benefit from metabolic considerations, although the authors repeatedly underline the
intention of highlighting perspective beyond physical measurements in
the assessment of the state of health.
In paragraph 4.2, among the limitations, one could add the impossibility
of evaluating some "organic mechanisms" at the basis of the improved
well-being produced by the change in lifestyle in the
absence of purely metabolic and bioclinical evaluations/measurements.

The importance of increased physical activity and fruit consumption on
"mental health" appears under-discussed from a metabolic point of view.

Round 2

Reviewer 1 Report

I'm pleased with the authors' answers and revision.